# Parental Anxiety, Practices, and Parent–Child Relationships among Families with Young Children in China: A Cross-Sectional Study

**DOI:** 10.3390/children10081388

**Published:** 2023-08-15

**Authors:** Wenya Yu, Zhichao Guo, Jiahe Tian, Panpan Li, Peng Wang, Hong Chen, Dan Zcm, Meina Li, Yang Ge, Xiang Liu

**Affiliations:** 1School of Public Health, Shanghai Jiao Tong University School of Medicine, Shanghai 200025, China; yuwenya@shsmu.edu.cn (W.Y.);; 2Department of Prevention and Health Care, Yuepu Town Community Health Service Center of Baoshan District, Shanghai 200941, China; 3Department of Prevention and Health Care, Dapuqiao Community Health Service Center of Huangpu District, Shanghai 200001, China; 4Department of Respiratory Disease, The 903rd Hospital of PLA, Hangzhou 310013, China; 5Department of Military Medical Services, College of Military Health Service, Naval Medical University, Shanghai 200433, China

**Keywords:** COVID-19, parenting, anxiety, parent–child activity, parent–child relationship

## Abstract

This study explored the ambiguous characteristics and influencing factors of parental anxiety, practices, and parent–child relationships among families with young children during a sudden COVID-19 lockdown in Shanghai, China. An online survey was conducted from 1 June to 10 November 2022, with 477 valid responses. Parental anxiety, practices, and parent–child relationships were evaluated. During this lockdown, 72.6% caregivers felt anxious about parenting to different degrees, with only a small proportion experiencing extreme anxiety. Parental anxiety was mainly influenced by whether the caregivers faced parenting issues and external parenting difficulties. The frequency of two-parent–child activities of reading books or looking at picture books with their children and telling stories to them significantly increased. Caregivers’ occupations of either professional or technical personnel and working from home were the most significant influencing factors. Mother–child relationships were relatively good. In conclusion, parental anxiety, practices, and parent–child relationships were relatively good and stable among families with young children during this lockdown. In the context of public health emergencies like COVID-19, more parenting support and knowledge should be provided to caregivers from professionals in CHCs or hospitals to decrease parental anxiety and improve parent–child relationships. Full advantage should be taken of working from home to promote parent–child activities.

## 1. Introduction

The global coronavirus disease 2019 (COVID-19) pandemic has caused significant social, economic, and medical threats to all people. By 22 December 2022, there were 650,879,143 confirmed cases and 6,651,415 deaths globally [1]. To mitigate the transmission of COVID-19, lockdown regulations were instituted in most countries [2,3], including China.

The national lockdown in China was implemented only during the first outbreak peak in Wuhan, in December 2019. Thereafter, routine prevention and control measures, such as wearing masks, have been adopted, provided there is no sudden large-scale outbreak or high-risk transmission. Many regions have not been significantly affected by COVID-19 after the first outbreak peak and have continued implementing routine prevention and control measures, which have minimum negative effects. Regions not significantly affected by COVID-19 included Shanghai before 2022, where people’s lifestyle and health remained relatively stable.

However, there was a sudden, large-scale outbreak of COVID-19 in March 2022 in Shanghai, which caused 57,595 confirmed cases and 581 deaths [4]. Owing to the rapid spread of COVID-19, the increasing demand for people to mitigate the transmission, and the important and special position of Shanghai, this outbreak led to a lockdown of the entire city for more than two months, from 28 March to 1 June 2022. Due to the sudden lockdown measures, people in Shanghai underwent a chaotic period of adjustment, during which vulnerable populations, including families with young children, might have faced more specific challenges [5,6].

Based on the Nurturing Care Framework [7], responsive caregiving theory [8], and ecosystem theory [9], interactions between caregivers and children are closely related to children’s developmental health, especially in the early stage of life; such interactions are mainly influenced by caregivers’ parental characteristics in the home environment [10]. Specifically, ecosystem theory posits that caregivers’ behaviors and parent–child relationships constitute the most crucial connections within the entire system [11], and the physical and mental health of caregivers constitutes the secondary links [12]. Therefore, caregivers’ parental characteristics are typically reflected as caregivers’ mental health, behaviors, and relationships with their children [13].

Previous research has indicated that, under the conditions of COVID-19 lockdowns, the characteristics and determinants of caregivers’ mental health, behaviors, and parent–child relationships become more complex than under normal circumstances. This may be attributable to the unique work and living conditions presented during the lockdown [14]. Specifically, caregivers faced great challenges to their mental health, reflected as a significant change in their parental anxiety levels [15,16] and greatly influenced by various factors, such as caregivers’ marital status, physical disability, economic hardship, and pre-existing mental health issues [17]. A study conducted in the Republic of Ireland and Italy further demonstrated the effects of COVID-19 lockdown on parental anxiety and explored key influencing factors [14]. Additionally, caregivers’ behaviors during lockdown were influenced by the way people worked. Most caregivers had to move their work from offices and other outside locations to their homes, which increased the time spent with children and might have further changed parent–child activities. However, working from home had two opposite effects on parental practices. On the positive side, working from home increased warmth and parent–child interactions [18,19]. On the negative side, it increased parent–child conflicts and reduced parent–child activities [20,21]. Furthermore, family and parent–child relationships during the COVID-19 lockdown may have changed significantly, and correlate with different factors, including family income, time spent together, the age and number of children, and caregivers’ workload [14,22].

Moreover, people were faced with unprecedented increases in the risk of infection and mortality rates during the large-scale lockdown in Shanghai, whereas simultaneously, some other countries and regions were gradually lifting lockdown measures. Thus, it is difficult to speculate on how the parental characteristics of caregivers with young children were affected during this period. Additionally, caregivers’ parental characteristics during the COVID-19 lockdown were much more special among families with young children. Compared to school-aged children and adolescents, young children have more care demands, are more dependent on their caregivers, and spend more time with their parents, making them more easily influenced by parental characteristics [23,24,25].

Considering the special features of young children and lack of localized evidence on the special stage of the lockdown in Shanghai, we hypothesized that parental anxiety, parental practices, and parent–child relationships among families with young children were significantly influenced by the caregivers’ and children’s characteristics and factors related to the COVID-19 lockdown. Therefore, this study aimed to (1) describe the characteristics of parental anxiety, parental practices, and parent–child relationships; (2) clarify the influence of children’s characteristics, family features, and experiences related to the pandemic on parental anxiety, parental behaviors, and parent–child relationships; and (3) explore the changes in parental practices among families with young children during the COVID-19 lockdown in Shanghai, China. This study was intended to provide evidence on minimizing the negative effects of pandemic prevention and control policies on families with young children from the perspective of parental characteristics.

## 2. Materials and Methods

### 2.1. Study Design

Considering the lockdown measures and the feasibility of participation, we designed an Internet-based survey to explore the changes in parental anxiety, parental practices, and parent–child relationships among families with young children during the COVID-19 lockdown in Shanghai, China. The required sample size was calculated to be at least 369, based on a confidence level of 95%, an admissible error of 0.1, and a prevalence rate of COVID-19 among young children of 2.0% [26]. To ensure the reliability and validity of this self-designed questionnaire, Cronbach’s α (0.786), Kaiser–Meyer–Olkin (KMO) test (0.890), and Bartlett’s test of sphericity (15,295.331, *p* < 0.0001) were calculated, which indicated both good internal consistency and validity.

The questionnaire included 22 items classified into 4 categories. The first category was the characteristics of the caregivers and families, with eight items, including the caregivers’ roles, sex, age, educational level, occupation, marital status, annual household income, and family residence.

The second category was the children’s features, with six items, including age, preterm birth, low birth weight, neonatal intensive care unit (NICU) admission, feeding patterns, and whether the child had siblings. If there were multiple children in a family, all information mentioned above should be about the youngest child of that family.

The third category was COVID-19-related information, with five items, including the number of family members living together during the lockdown, whether someone they knew was infected by COVID-19 during the lockdown, whether they worked from home during the lockdown, whether they faced parenting issues, and whether they faced external parenting difficulties.

The fourth category was three parental characteristics closely related to children’s developmental health based on the Nurturing Care Framework and responsive caregiving theory, including parental anxiety, parental practices, and parent–child relationships. First, parental anxiety was used to reflect the caregivers’ mental health status. The caregivers were asked to self-report their level of parental anxiety based on a 5-point Likert scale [27]. Second, parental practices reflected as five activities were used to describe the caregivers’ behaviors, which were proposed by the Multiple Indicator Cluster Surveys 6 (MICS6) Questionnaire for Children Under Five [28]. The five activities included the frequency of reading books or looking at picture books with their children, telling stories to them, singing songs to or with their children (including lullabies), playing with them, and naming, counting, or drawing things for or with them. We requested that the caregivers reported the frequency of these activities one month prior to and during the COVID-19 lockdown. Additionally, since it was impossible to play outside the home during the lockdown, we excluded one activity in MICS6, namely, taking a child outside the home. Third, considering the difficulty of measuring the feelings about parent–child relationships of children under three years of age, and the importance of mothers in caring for children at that age, we evaluated parent–child relationships based on the Mother’s Object Relations Scales-Short Form (MORS-SF) [29]. The MORS-SF assesses mothers’ perceptions of their child on a scale of warmth and invasiveness. If a mother feels unduly invasive or less warm, the parent–child relationship is not satisfactory. Mothers were required to assess their feelings regarding seven items of both warmth and invasiveness based on a 6-point scale.

### 2.2. Data Collection

Because the overall lockdown was lifted on 1 June 2022, and the new national COVID-19 prevention and control policy changed significantly on 11 November 2022, the formal investigation for this study was conducted from 1 June to 10 November 2022, to ensure that all participants underwent the complete lockdown period and completed the questionnaire under the same COVID-19 prevention and control policy. Considering children aged 0–3 years are required to have regular check-ups with childcare professionals at community health service centers (CHCs), their parents are likely to have a good level of trust in these childcare professionals. Therefore, childcare providers at CHCs in 11 districts of Shanghai helped to disseminate the questionnaires to ensure the good quality of the survey. We recruited parents with at least one child aged 0–3 years from the childcare databases of CHCs in 11 districts of Shanghai. Using a simple random sampling method, 552 individuals were chosen, and questionnaires were distributed to them online. If the person registered in the database was not the actual caregiver of the child, the questionnaire would be handed over to the actual caregiver for completion. Based on an integrity and reasonableness check, 477 responses were valid without missing data (valid response rate = 86.4%).

The inclusion criteria were as follows: (1) caregivers should have at least one child under three years; (2) caregivers should live with their children under three years in Shanghai from at least 28 March to 1 June 2022; (3) caregivers should voluntarily participate in this survey; and (4) participating caregivers should complete the investigation and provide informed consent online. The exclusion criteria were (1) caregivers with a child under three years who did not live in Shanghai during lockdown; and (2) caregivers who could not complete the online survey and provide informed consent.

### 2.3. Statistical Analysis

Statistical analyses were conducted using PASW Statistics for Windows (version 18.0, SPSS Inc., Chicago, IL, USA) and SAS 8.2 (SAS Institute Inc., Cary, NC, USA). Descriptive statistics were employed to describe the basic characteristics of caregivers, children, COVID-19 related information, and parental characteristics. Comparisons of the frequency of parental practices before and during the COVID-19 lockdown were examined using the Wilcoxon rank test owing to the ordinal classification type of these variables. Only parental practices with significant changes were further analyzed with univariate and multivariate analyses. Univariate analysis used nonparametric tests, based on the ordinal classification type of the three parental characteristics, to assess the relationships between the dependent variables and each independent variable. The Wilcoxon rank test was used for binary independent variables, and the Kruskal–Wallis H test for multicategory independent variables. Variables with statistical significance were included in stepwise logistic regression analysis or stepwise multiple linear regression analysis, which were conducted to analyze the factors influencing parental anxiety, parental practices, and parent–child relationships. In terms of the dependent variable of parental anxiety, it was measured on a scale ranging from “none” to “extremely high” and determined as an ordinal variable. Considering parental practices, based on the survey results, the monthly frequency of parent–child activities was assigned a value from 0 (never) to 6 (everyday). The difference between the values pre- and post-pandemic was then calculated as the dependent variable. Parent–child relationships were measured and assigned a value based on a 6-point warmth and invasiveness scale. Considering the dependent variable of mothers’ perceptions of their warmth towards their children, it was classified into three ordinal levels (0–11, 12–19, 20–35); the dependent variable of mothers’ perceptions of their invasiveness towards their children was also divided into three ordinal levels (0–12, 13–17, 18–35). Additionally, principal component analysis was conducted using the regression scores approach to address the multicollinearity. This method allowed the composite variable to reflect the severity of parenting issues and external parenting difficulties through a numerical magnitude [30]. All tests were two-tailed, and *p* < 0.05 was considered statistically significant.

### 2.4. Ethics Approval

Approval was obtained from the Ethics Committee of Shanghai Jiao Tong University School of Medicine School of Public Health (SJUPN-202109). All the participants provided informed consent. The procedures used in this study adhered to the tenets of the Declaration of Helsinki.

## 3. Results

### 3.1. Demographic, Parental, and Lockdown-Related Characteristics of Caregivers, Families, and Children

Table 1 shows the demographic, parental, and lockdown-related characteristics of the caregivers, families, and children. The majority of caregivers who participated were mothers (93.1%) and married. The average age was 32 years, and the largest proportion of caregivers were college graduates and office workers. The majority were from families with an annual household income between 100,000 and 200,000 CNY and who lived in the Baoshan district. The average age of children was 18 months, and approximately one-third had siblings; 5.5% of the children were born prematurely, 4.4% had a low birth weight, and 16.6% were admitted to the NICU; most children were formula-fed babies.

### 3.2. COVID-19-Related Information of Participants

During the COVID-19 lockdown in Shanghai, most of the participants reported to have four family members living together; 26.8% reported that someone they knew had a confirmed case of COVID-19; more than two-thirds were working from home; and parenting issues of child feeding and external parenting difficulties of fear of children being infected were most caregivers’ concerns (Table 1).

### 3.3. Characteristics of Parental Anxiety and its Influencing Factors

Overall, 72.6% of caregivers reported that they felt anxious about parenting to different degrees; however, only 14.7% felt relatively high or very high degrees of anxiety (Table 1).

The univariate analysis results (Appendix A) suggest that the caregivers’ parental anxiety was influenced by the children’s ages; the children’s feeding patterns; whether the caregivers knew someone infected by COVID-19; whether the caregivers faced child feeding issues, physical or other developmental issues, parent–child interaction issues; and whether the caregivers faced external parenting difficulties in accessing childcare providers in CHCs and pediatricians in hospitals (all *p*-values < 0.05).

The logistic regression analysis results (Table 2) indicate that the caregivers’ parental anxiety was influenced by whether they faced parenting issues or external parenting difficulties. Based on principal component analysis, feeding the child, physical and other aspects of development, and parent–child interaction issues were extracted as one factor, referred to as “facing parenting issues”. This factor has eight classifications and is represented by numbers; a higher number indicates more parenting issues faced by the caregivers. Difficulties in accessing childcare providers in CHCs and pediatricians in hospitals were extracted as one factor, referred to as “external parenting difficulties”. This factor has three classifications, and a larger number suggests more external parenting difficulties faced by the caregivers. Specifically, caregivers with lower scores for the factors of facing parenting issues and external parenting difficulties were less likely to have high levels of parental anxiety.

### 3.4. Changes in Parental Practices and their Influencing Factors

Significant increases in the frequency of reading books or looking at picture books with children and telling stories to them were observed during the COVID-19 lockdown. Changes in the frequencies of the other three parent–child activities were not statistically significant (Table 3).

A univariate analysis of significant changes in the frequencies of parent–child activities was conducted. Changes in the frequency of reading books or looking at picture books with their children were significantly influenced by the caregivers’ occupations, family residence, children’s age, whether the caregivers worked from home, and whether the caregivers were anxious about their children’s physical development (Appendix A). The above-mentioned variables were included in the regression analysis. The regression analysis results (Table 4) show that having a family residence in the Minhang, Baoshan, and Pudong districts; children being aged 3–6 months; and caregivers working from home were positively associated with an increased frequency of reading books or looking at picture books with children. However, caregivers’ occupations of either professional or technical personnel; cadres of political parties and government organs and institutions, civil servants, village, and neighborhood committee workers; other occupations; and children being aged 30–36 months were negatively associated with increased frequency.

Changes in the frequency of telling stories to one’s child were significantly influenced by caregivers’ age, caregivers’ occupations, family residence, whether caregivers worked from home, and anxiety about the child’s physical development (Appendix A). The above-mentioned variables were included in the regression analysis. The regression analysis results (Table 5) suggested that caregivers working from home was positively associated with an increased frequency in telling stories to children. However, caregivers’ occupation of professional and technical personnel was negatively associated with frequency increase.

### 3.5. Characteristics of Parent–Child Relationships and Influencing Factors

Of the 444 enrolled mothers, the average scores for their perceptions of warmth and invasiveness towards their children were 31 and 7, respectively, suggesting that mother–child relationships were relatively good during the COVID-19 lockdown (Table 1).

Regarding parent–child relationships, the univariate analysis suggests that none of the factors were statistically significant (Appendix A). The regression analysis was not conducted according to the univariate analysis.

## 4. Discussion

This study explored the ambiguous characteristics and influencing factors of parental anxiety, parental practices, and parent–child relationships among families with young children during the COVID-19 lockdown in Shanghai, China. Our findings indicate that 72.6% of caregivers experienced parental anxiety to a variable magnitude, which was mainly influenced by whether they faced parenting issues or external parenting difficulties. The frequencies of two-parent–child activities of reading books or looking at picture books with children and telling stories to them significantly increased. The most significant influencing factors were caregivers’ occupations of professional and technical personnel and whether the caregivers worked from home. From the mothers’ perceptions, parent–child relationships were relatively good during the COVID-19 lockdown.

In the context of a large-scale COVID-19 lockdown in a megacity of China, characteristics of parental anxiety, practices, and parent–child relationships among families with young children were clarified for the first time. This study provides much needed information for other countries or regions to conduct better interventions to maintain and accelerate parenting-related issues among families with young children during lockdowns for public health emergencies, such as the sudden COVID-19 lockdown. Different from findings on the influence of the COVID-19 lockdown on parental characteristics among families with school-aged children and adolescents, several encouraging results about families with young children were found in this study.

The overall degree of parental anxiety of caregivers with young children in Shanghai was relatively low, with only 3.6% feeling extremely anxious, which was much lower than that reported in data from other countries [16]. However, data reported elsewhere were usually based on children within a wider age range, indicating that parental anxiety from COVID-19 lockdown for families with young children was not as serious as we thought. This may be attributed to the less dramatic lifestyle changes in young children. For example, older children needed to take online lessons at home, which required caregivers to perform more duties that had previously been provided by schools and teachers, which significantly increased caregivers’ parenting burden [14,31]. Therefore, it could be understandable that caregivers with young children were more emotionally stable than parents of other children because their lifestyles had not changed drastically. To further explore the influencing factors of parental anxiety in this study, the reason for the lower degree of anxiety may be attributed to fewer parenting issues and external parenting difficulties. The lockdown in Shanghai occurred in the later stage of the COVID-19 pandemic; therefore, caregivers in this stage had gained more prevention, control, and treatment knowledge than those in the early stage. This helped them to experience fewer parenting issues and difficulties, which could reduce their parental anxiety [5]. Additionally, consistent with other studies, parenting issues referring to daily caregiving and child development issues [20,32], issues related to parent–child interactions [33,34,35,36,37], and childcare resource access [20,32] were key factors influencing caregivers’ parental anxiety.

The parent–child relationships in families with young children were relatively good during the sudden COVID-19 lockdown in Shanghai, based on the perspective of mothers. This phenomenon was closely related to the low degree of anxiety. A lower degree of caregiver anxiety would lead to better parent–child relationships [38,39] by improving caregivers attitudes towards their children and the quality of the relationship [40]. Although findings about parent–child relationships in other countries, such as the Netherlands [41] and the United States [20], were contradictory to our findings in that the relationships became worse during the COVID-19 lockdown, they mainly concentrated on school-aged children or adolescents rather than young children. Therefore, the good parent–child relationships between caregivers and children under three years might be attributed to children’s features at this age. Different from school-aged children or adolescents, who would have more conflicts with caregivers due to changes in lifestyles, behaviors, physical activities, media usage, and online learning [20,42,43], it was less likely for young children under three years to have conflicts with caregivers. However, it has been widely recognized that parent–child conflicts are key in deteriorating parent–child relationships [20]. Thus, fewer parent–child conflicts protected parent–child relationships.

The frequencies of parent–child activities of reading books, looking at picture books with children, and telling stories to them significantly increased among families with young children. This phenomenon was also observed in other countries, where caregivers spent more time accompanying their children to parent–child activities [18,19,44,45]. The main reason for this positive change may be partly attributed to caregivers working from home, which could increase family time. However, there should be additional focus on families where caregivers were either professional or technical personnel, because the frequency of parent–child activities decreased in these families. Most caregivers in this profession were healthcare providers and teachers, which greatly limited their family time during the COVID-19 lockdown. Almost all healthcare providers participated in pandemic prevention and control activities, and it was impossible to be at home with their children. For teachers, providing online courses greatly reduced the time they could have spent with their children.

This study has several limitations. First, owing to the infeasibility of measuring young children’s feelings about parent–child relationships, these relationships were only evaluated from the perspective of mothers. Second, recall bias may exist, since participants were required to report parent–child activities before the COVID-19 lockdown. Third, a potential selection bias may have influenced the results, as this was an online survey, although we invited childcare providers in different districts of Shanghai to expand the scope of this investigation to minimize such bias. Fourth, some family factors, such as religion, were not measured. Face-to-face in-depth interviews should be conducted to yield better results. Fifth, the frequency of parent–child activities during and prior to the pandemic might have been affected by the significant physical and intellectual developments that typically occur in children aged 0–3 years. For example, if a child was 4 months old prior to the lockdown, this child would have been 6 months old during the lockdown, meaning that the caregiver should provide responses regarding the frequency of reading books to a child at 4 and 6 months of age. Although we considered two months to be a relatively short period with small developmental changes, older children usually read more books, which might have introduced some bias. Sixth, the identified characteristics and influencing factors of parental anxiety, parental practices, and parent–child relationships among families with young children in this study were highly dependent on the context of the pandemic lockdown. Thus, the universality in maintaining and accelerating parenting issues during public health emergencies should be further explored.

## 5. Conclusions

The overall parental characteristics were relatively good and stable among families with young children in Shanghai, China, which protected young children’s developmental health during the COVID-19 lockdown. The degree of parental anxiety was low, the frequencies of parent–child activities increased, and parent–child relationships were good. It may help decrease parental anxiety if parenting issues and difficulties are reduced by caregivers receiving more parenting support and knowledge from professionals in CHCs or hospitals. Decreased parental anxiety would also help promote parent–child relationships. We encouraged caregivers to take advantage of working from home to improve parent–child activities; however, more social support should be provided to caregivers in professional and technical occupations to protect their ability to engage in effective parent–child activities in these families during special periods.

## Figures and Tables

**Table 1 children-10-01388-t001:** Demographic, parental, and lockdown-related characteristics.

Characteristic	*N*/Medium	%/IQR
Caregiver’s role		
Father	25	5.2
Mother	444	93.1
Grandparents	6	1.3
Other guardian	2	0.4
Caregiver’s sex		
Male	29	6.1
Female	448	93.9
Caregiver’s age (years)	32	30–35
20–29	113	23.7
30–39	333	69.8
40–49	29	6.1
≥50	2	0.4
Caregiver’s educational level		
Junior high school degree	14	2.9
Junior college degree	45	9.4
Bachelor’s degree	375	78.6
Master’s degree	37	7.8
Doctorate degree	6	1.3
Caregiver’s occupation		
Worker (e.g., factory workers/manual laborers)	14	2.9
Office worker	194	40.7
Business service personnel (e.g., waiters, salesman, drivers)	6	1.3
Professional and technical personnel (e.g., health care providers, teachers)	122	25.6
Cadres of party and government organizations and institutions, civil servants, village, and neighborhood committee workers	13	2.7
Managers of state-owned enterprises (including middle and grass-roots managers)	8	1.7
Managers of private and foreign-funded enterprises (including middle-level and grass-roots managers)	13	2.7
Individual industrial and commercial enterprises	10	2.1
Freelancers	40	8.4
Agricultural laborers	1	0.2
Unemployed	45	9.4
Others	11	2.3
Caregiver’s marital status		
Single	1	0.2
Married	475	99.6
Divorced	1	0.2
Annual household income (CNY)		
<100,000	74	15.5
(100,000, 200,000)	191	40.0
(200,000, 300,000)	107	22.4
(300,000, 500,000)	79	16.6
(500,000, 800,000)	18	3.8
(800,000, 1,000,000)	4	0.8
≥1,000,000	4	0.8
Family residence (district)		
Huangpu	42	8.8
Xuhui	5	1.0
Putuo	1	0.2
Hongkou	1	0.2
Yangpu	15	3.1
Minhang	19	4.0
Baoshan	374	78.4
Jiading	12	2.5
Pudong new area	3	0.6
Jinshan	3	0.6
Songjiang	2	0.4
Whether the youngest child has siblings		
Yes	141	29.6
No	336	70.4
Age of the youngest child (months)	18	10–29
≤1	11	2.3
(1, 3)	2	0.4
(3, 6)	25	5.2
(6, 8)	39	8.2
(8, 12)	95	19.9
(12, 18)	70	14.7
(18, 24)	74	15.5
(24, 30)	64	13.4
(30, 36)	97	20.3
Whether the youngest child was born prematurely (<37 weeks)		
Yes	26	5.5
No	451	94.5
Whether the youngest child was born with low birth weight (<2500 kg)		
Yes	21	4.4
No	456	95.6
Whether the youngest child was admitted to NICU		
Yes	79	16.6
No	398	83.4
Feeding pattern of the youngest child		
Breast feeding	78	16.4
Mixed feeding	186	39.0
Formula feeding	213	44.7
Number of family members living together during lockdown		
2	23	4.8
3	105	22.0
4	161	33.8
5	139	29.1
6	39	8.2
7	6	1.3
8	4	0.8
Whether someone you know was infected by COVID-19 during lockdown		
Yes	128	26.8
No	349	73.2
Whether you worked from home during lockdown		
Yes	331	69.4
No	146	30.6
Whether you faced the following parenting issues		
Child feeding (e.g., nutrition, breastfeeding, food types)	198	41.5
Child physical development (e.g., height, weight)	192	40.3
Child other development (e.g., gross motor, language, social-emotional)	166	34.8
Parent–child interaction (e.g., how to accompany a child, how to educate a child)	195	40.9
Other	23	4.8
Whether you faced the following external parenting difficulties		
Difficulty in accessing professional guidance from childcare providers in CHCs	188	39.4
Difficulty in accessing professional guidance from pediatricians in hospitals	174	36.5
Fear of child being infected	231	48.4
Other	36	7.5
Level of parenting anxiety		
None	107	22.4
A little bit	276	57.9
Uncertain	24	5.0
Relatively high	53	11.1
Extremely high	17	3.6
Mother’s perceptions of warmth towards her child	31	23–35
0–11	15	3.1
12–19	93	19.5
20–35	336	70.4
Mother’s perceptions of invasiveness towards her child	7	5–11
0–12	393	82.4
13–17	33	6.9
18–35	18	3.8

**Table 2 children-10-01388-t002:** Influencing factors of parental anxiety during the COVID-19 lockdown.

Characteristic	Estimate	Wald Chi-Square	*p*-Value	OR	95% Wald Confidence Limits	
					Lower	Upper
Facing parenting issues						
0.26	−1.431	3.190	0.074	0.239	0.050	1.150
0.40	−0.950	6.673	0.010 *	0.387	0.188	0.795
0.41	−1.547	8.884	0.003 *	0.213	0.077	0.589
0.55	−0.940	6.354	0.012 *	0.391	0.188	0.811
0.56	−1.314	9.023	0.003 *	0.269	0.114	0.633
0.70	−0.472	0.829	0.363	0.624	0.226	1.723
0.71	−0.831	4.342	0.037 *	0.436	0.200	0.952
0.85	Ref	Ref	Ref	Ref	Ref	Ref
Facing external parenting difficulties						
0.27	−0.632	4.218	0.040 *	0.532	0.291	0.972
0.57	−0.259	0.852	0.356	0.772	0.446	1.337
0.88	Ref	Ref	Ref	Ref	Ref	Ref

* Indicates statistically significant results (*p* < 0.05).

**Table 3 children-10-01388-t003:** Frequencies of parent–child activities before and during this COVID-19 lockdown.

Characteristic	Before Lockdown		During Lockdown		Mann–Whitney U	*p*-Value
	*N*	%	*N*	%		
Frequency of reading books or looking at picture books with child					104,366	0.018 *
Everyday	200	41.9	252	52.8		
Once every 2–3 days	146	30.6	102	21.4		
Once every 4–5 days	42	8.8	33	6.9		
Once every 6–7 days	24	5.0	20	4.2		
Once every two weeks	13	2.7	14	2.9		
Once a month	21	4.4	19	4.0		
Never	31	6.5	37	7.8		
Frequency of telling stories to child					101,908	0.003 *
Everyday	186	39.0	245	51.4		
Once every 2–3 days	136	28.5	98	20.5		
Once every 4–5 days	41	8.6	33	6.9		
Once every 6–7 days	29	6.1	22	4.6		
Once every two weeks	25	5.2	14	2.9		
Once a month	22	4.6	27	5.7		
Never	38	8.0	38	8.0		
Frequency of singing songs to or with child (including lullabies)					109,902	0.295
Everyday	287	60.2	308	64.6		
Once every 2–3 days	92	19.3	75	15.7		
Once every 4–5 days	31	6.5	23	4.8		
Once every 6–7 days	24	5.0	23	4.8		
Once every two weeks	13	2.7	8	1.7		
Once a month	10	2.1	17	3.6		
Never	20	4.2	23	4.8		
Frequency of playing with child					113,259	0.865
Everyday	381	79.9	381	79.9		
Once every 2–3 days	46	9.6	36	7.5		
Once every 4–5 days	12	2.5	14	2.9		
Once every 6–7 days	15	3.1	13	2.7		
Once every two weeks	3	0.6	6	1.3		
Once a month	2	0.4	9	1.9		
Never	18	3.8	18	3.8		
Frequency of naming, counting, or drawing things for or with child					108,035	0.114
Everyday	293	61.4	323	67.7		
Once every 2–3 days	100	21.0	71	14.9		
Once every 4–5 days	24	5.0	20	4.2		
Once every 6–7 days	26	5.5	19	4.0		
Once every two weeks	3	0.6	6	1.3		
Once a month	8	1.7	17	3.6		
Never	23	4.8	21	4.4		

* Indicates statistically significant results (*p* < 0.05).

**Table 4 children-10-01388-t004:** Influencing factors of the change in frequency of reading books or looking at picture books with children owing to the COVID-19 lockdown.

Characteristic	b	Std. Error	b′	t	*p*-Value
Caregiver’s occupation					
Other	−0.932	0.469	−0.088	−1.990	0.048 *
Professional and technical personnel	−0.341	0.167	−0.094	−2.040	0.042 *
Cadre of political parties and government organizations and institutions, civil servants, village, and neighborhood committee workers	−0.897	0.429	−0.092	−2.090	0.037 *
Worker	Ref	Ref	Ref	Ref	Ref
Family residence					
Yangpu	0.817	0.439	0.090	1.860	0.064
Minhang	0.974	0.400	0.120	2.440	0.015 *
Baoshan	0.759	0.210	0.197	3.610	<0.001 *
Pudong new area	1.887	0.887	0.094	2.130	0.034 *
Huangpu	Ref	Ref	Ref	Ref	Ref
Age of the youngest child (months)					
(30, 360	−0.580	0.178	−0.147	−3.250	0.001 *
(3, 6)	0.745	0.314	0.104	2.370	0.018 *
(18, 24)	−0.349	0.197	−0.080	−1.770	0.077
≤1	Ref	Ref	Ref	Ref	Ref
Working from home					
Yes	0.411	0.155	0.119	2.650	0.008 *
No	Ref	Ref	Ref	Ref	Ref

* Indicates statistically significant results (*p* < 0.05).

**Table 5 children-10-01388-t005:** Influencing factors of the change in frequency of telling stories to children owing to the COVID-19 lockdown.

Characteristic	b	Std. Error	b′	t	*p*-Value
Caregiver’s age					
40–49	−0.473	0.285	−0.074	−1.660	0.098
20–29	Ref	Ref	Ref	Ref	Ref
Caregiver’s occupation					
Other	−0.875	0.453	−0.086	−1.930	0.054
Professional and technical personnel	−0.406	0.164	−0.116	−2.470	0.014 *
Cadres of political parties and government organizations and institutions, civil servants, village, and neighborhood committee workers	−0.694	0.419	−0.074	−1.660	0.098
Worker	Ref	Ref	Ref	Ref	Ref
Family residence					
Songjiang	−1.929	1.044	−0.081	−1.850	0.065
Xuhui	−1.250	0.682	−0.083	−1.830	0.068
Baoshan	0.274	0.181	0.074	1.520	0.130
Jiading	−0.873	0.472	−0.089	−1.850	0.065
Huangpu	Ref	Ref	Ref	Ref	Ref
Working from home					
Yes	0.434	0.151	0.130	2.860	0.004 *
No	Ref	Ref	Ref	Ref	Ref
Facing parenting issue of child physical development					
Yes	0.202	0.137	0.065	1.480	0.141
No	Ref	Ref	Ref	Ref	Ref

* Indicates statistically significant results (*p* < 0.05).

## Data Availability

The data presented in this study are available upon request from the corresponding author. The data are not publicly available due to ethical restrictions.

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
