# Peer review of "Parental Anxiety, Practices, and Parent–Child Relationships among Families with Young Children in China: A Cross-Sectional Study"

_children, 2023, doi:10.3390/children10081388_

Round 1

Reviewer 1 Report

·       There is not enough information about how the sample was collected and from what population. It is difficult to tell how representative (and thus how legitimate the inferential statistics are) the presented data is.

·       I do not understand what the first column in Table 3 is

·       The dependent variables for the tables are not clearly explained. For example, in Table 4, how is the dependent variable measured?

·       Related, the reader needs more information on the structure of the questions that generated the independent variables. The authors asked people to report on extensive scales before lockdown – how long ago was that? The lockdown ended June 1 – when did it begin? Some of the people interviewed might not have a had children. All of them would have had children at a significantly different developmental stage. For example, if lockdown was a year, a parent would be answering about a 6month old and then an 18 month old; of course they read them more books.

·       Related, in the discussion, the authors say “This study explored the ambiguous effects of the sudden COVID-19 lockdown on families with young children in Shanghai, China.” But, again, data were not collected before and after the lockdown. How can the authors claim to be studying the effect of the lockdown?

·       The conclusion says the results could provide “much-needed information for other countries or regions to conduct better parenting-related interventions to maintain and accelerate young children’s developmental health during public health emergencies such as the sudden COVID-19 lockdown.” I think such a result would be highly context dependent, especially with respect to the terms of the lockdown.

Minor editing required.

Author Response

Question 1: There is not enough information about how the sample was collected and from what population. It is difficult to tell how representative (and thus how legitimate the inferential statistics are) the presented data is.

Response: Thank you for your valuable suggestions. As for sample information and collection process, we have added more detailed descriptions as follows: “Considering children aged 0–3 years are required to have regular check-ups with child-care professionals in community health service centers (CHCs), their parents are likely have a good level of trust in these childcare professionals. Therefore, childcare providers at CHCs in 11 districts of Shanghai helped to disseminate the questionnaires to ensure the good quality of the survey. We recruited parents with at least one child aged 0–3 years from the childcare databases of CHCs in 11 districts of Shanghai. Using a simple random sampling method, 552 individuals were chosen, and questionnaires were distributed to them online. If the person registered in the database was not the actual caregiver of the child, the questionnaire would be handed over to the actual caregiver for completion.” Please refer to pp. 4, “2.2. Data Collection”.

Question 2: I do not understand what the first column in Table 3 is.

Response: When analyzing the relationship between parental anxiety levels and parenting issues, as well as external parenting challenges, we performed multicollinearity tests separately for the four types of caregiving issues and the two external parenting challenges. The variance inflation factors (VIF) were all found to be greater than 10, indicating multicollinearity. Consequently, in the regression model calculations, we employed the principal component analysis (PCA) method to consolidate the four types of caregiving issues into a single caregiving factor and the two external parenting challenges into a single external parenting factor, represented by numeric values. We provided an explanation for the use of the principal component analysis (PCA) method, as follows: “Additionally, principal component analysis was conducted using the regression scores approach to address the multicollinearity. This method allowed the composite variable to reflect the severity of parenting issues and external parenting difficulties through a numerical magnitude [30].” Please refer to pp. 4, “2.3. Statistical Analysis”.

Furthermore, in the Results section, we adjusted the order of tables, renumbering the original Table 3 as Table 2, and provided a detailed explanation for the numbers in its first column as follows: “Based on principal component analysis, feeding the child, physical and other aspects of development, and parent-child interaction issues were extracted as one factor, referred to as “facing parenting issues.” This factor has eight classifications and is represented by numbers; a higher number indicates more parenting issues faced by the caregivers. Difficulties in accessing childcare providers in CHCs and pediatricians in hospitals were extracted as one factor, referred to as “external parenting difficulties.” This factor has three classifications, and a larger number suggests more external parenting difficulties faced by the caregivers.” Please refer to pp. 7, “3.3. Characteristics of parental anxiety and its influencing factors”.

Question 3: The dependent variables for the tables are not clearly explained. For example, in Table 4, how is the dependent variable measured?

Response: Thank you. We have added a statement for clearer explanations of dependent variables and measurement as follows: “In terms of the dependent variable of parental anxiety, it was measured on a scale ranging from “none” to “extremely high” and determined as an ordinal variable. Considering pa-rental practices, based on the survey results, the monthly frequency of parent-child activi-ties was assigned a value from 0 (never) to 6 (everyday). The difference between the values pre- and post-pandemic was then calculated as the dependent variable. Parent-child rela-tionships were measured and assigned a value based on a 6-point warmth and invasive-ness scale. Considering the dependent variable of  mothers’ perceptions of their warmth towards their children, it was classified into three ordinal levels (0–11, 12–19, 20–35); the dependent variable of mothers’ perceptions of their invasiveness towards their children was also divided into three ordinal levels (0–12, 13–17, 18–35).” Please refer to pp. 4, “2.3. Statistical Analysis”.

Question 4: Related, the reader needs more information on the structure of the questions that generated the independent variables. The authors asked people to report on extensive scales before lockdown – how long ago was that? The lockdown ended June 1 – when did it begin? Some of the people interviewed might not have a had children. All of them would have had children at a significantly different developmental stage. For example, if lockdown was a year, a parent would be answering about a 6month old and then an 18 month old; of course they read them more books.

Response: First, when inquiring about the frequency of parent-child activities prior to the pandemic lockdown, we defined it as the average value during the month before the lockdown, which was clearly clarified in the questionnaire. Please refer to pp. 3, “2.1. Study Design”, as follows: “We requested that caregivers report the frequency of these activities one month prior to and during the COVID-19 lockdown.”

Second, in terms of the pandemic lockdown timeline, we have provided a detailed statement in the Introduction section as follows: “this outbreak led to a lockdown of the entire city for more than two-months, from March 28 to June 1, 2022.” Please refer to pp. 2, “1. Introduction”.

Third, all respondents were drawn from newborn follow-up records at various community health service centers in Shanghai, ensuring that all participants had at least one child. Additionally, to ensure the accuracy of the questionnaire, if a parent listed in the records is not the primary caregiver, we would request to pass on the questionnaire to the actual caregiver. Please refer to pp. 4, “2.2. Data Collection”.

Fourth, although the lockdown measures lasted for just over two months city-wide, we do agree that due to the rapid physical and intellectual development of children aged 0-3 years, a span of a few months could indeed influence the frequency of specific parent-child activities, such as increased reading or cartoon-watching time. We have added this as a limitation as follows: “Fifth, the frequency of parent-child activities during and prior to the pandemic might have been affected by the significant physical and intellectual developments that typically occur in children aged 0–3 years. For example, if a child was 4 months old prior to the lockdown, this child would have been 6 months old during the lockdown, meaning that the caregiver should provide responses regarding the frequency of reading books to a child at 4 and 6 months of age. Although we considered two months to be a relatively short period with small developmental changes, older children usually read more books, which might have introduced some bias.” Please refer to pp. 12, “4. Discussion”.

Question 5: Related, in the discussion, the authors say “This study explored the ambiguous effects of the sudden COVID-19 lockdown on families with young children in Shanghai, China.” But, again, data were not collected before and after the lockdown. How can the authors claim to be studying the effect of the lockdown?

Response: Apologize for the inappropriate statement. During the COVID-19 lockdown, many families experienced unprecedented difficulties, including food shortages, reduced income, and psychological stress due to the lockdown measurements. Considering the increase of potential bias in asking respondents to recall their pre-lockdown parenting experiences, this study only requested participants to report their parenting experiences during the lockdown, including parent-child relationships and parental anxiety. Consequently, it was impossible to explore changes in parent-child relationships and parental anxiety before and after the lockdown.

However, as for parent-child activities, since we only requested a report for the month before the lockdown, the relevant assessments by experts and pre-experiments indicated that the memory bias was relatively controllable. Therefore, based on caregivers' self-reports, we explored changes in parent-child activities before and during the lockdown.

In light of your advice, we have revised the title as “Parental Anxiety, Practices, and Parent-Child Relationships Among Families with Young children in China: A Cross-Sectional Study” for more accurate representation. Detailed revisions on clear statement and corresponding background and study design have also been made in the main text, please refer to pp. 2-3, “1. Introduction”. Further, we have restructured the research objectives as follows: “Therefore, this study aimed to (1) describe the characteristics of parental anxiety, parental practices, and parent-child relationships; (2) clarify the influence of children's characteristics, family features, and experiences related to the pandemic on parental anxiety, parental behaviors, and parent-child relationships; and (3) explore the changes in parental practices among families with young children during the COVID-19 lockdown in Shanghai, China. This study was intended to provide evidence on minimizing the negative effects of pandemic prevention and control policies on families with young children from the per-spective of parental characteristics.” We have also checked the whole manuscript and revised relevant statements in Abstract and Discussion.

Question 6: The conclusion says the results could provide “much-needed information for other countries or regions to conduct better parenting-related interventions to maintain and accelerate young children’s developmental health during public health emergencies such as the sudden COVID-19 lockdown.” I think such a result would be highly context dependent, especially with respect to the terms of the lockdown.

Response: Thank you and we do agree with you. Combined with the revised research objectives, we revised this statement as follows: “This study provides much-needed information for other countries or regions to conduct better interventions to maintain and accelerate parenting-related issues among families with young children during lockdowns for public health emergencies, such as the sudden COVID-19 lockdown.” Please refer to pp. 10-11, “4. Discussion”.

Additionally, it should also be a limitation of this study, which has been added as the sixth limitation, as follows: “Sixth, the identified characteristics and influencing factors of parental anxiety, parental practices, and parent-child relationships among families with young children in this study were highly dependent on the context of the pandemic lockdown. Thus, the universality in maintaining and accelerating parenting issues during public health emergencies should be further explored.” Please refer to pp. 12, “4. Discussion”.

Reviewer 2 Report

This is an interesting paper that can add new knowledge to the study of family dynamics and parents' wellbeing during Covid-19 pandemic. My comments are listed below.

Introduction:

-       I suggest that the authors consult this work and reference it since it is about the impact on Covid-19 on parents’ wellbeing during lockdown:

Everri, M., Messena, M., Nearchou, F., & Fruggeri, L. (2022). Parent-Child Relationships, Digital Media Use and Parents' Well-Being during COVID-19 Home Confinement: The Role of Family Resilience. International journal of environmental research and public health19(23), 15687. https://doi.org/10.3390/ijerph192315687

Despite the study being conducted in other countries, Everri et al. can provide additional information to discuss and compare with the results of this paper.

Materials and methods

-       Aim and research objectives should be specified. The way in which goals are presented fit a qualitative methodology, while this is a quantitative paper in which it is expected that authors have hypotheses and assumptions about the relationship among the considered variables.

-       Additionally, they should provide more details about the model that they planned to test.

-       Measures: please reference the scales used or specify if the scales were specifically developed for the study.

Results

-       The presentation of results would benefit from a better organisation of the material. This might be due to the lack of clarity in the study hypothesis and a clearer description of the considered variables.

-       There is reference to Table S1 that should be inserted.

-       I suggest that authors replace ‘basic characteristics’ with ‘demographic characteristics’.  

Overall, the paper requires major revisions. Perhaps authors should consider reducing the number of variables considered and develop stronger hypothesis. The paper lacks in consistency and this reduce the potential strengths of the results.

It is good.

Author Response

Question 1: I suggest that the authors consult this work and reference it since it is about the impact on Covid-19 on parents’ wellbeing during lockdown:

Everri, M., Messena, M., Nearchou, F., & Fruggeri, L. (2022). Parent-Child Relationships, Digital Media Use and Parents' Well-Being during COVID-19 Home Confinement: The Role of Family Resilience. International journal of environmental research and public health, 19(23), 15687. https://doi.org/10.3390/ijerph192315687

Despite the study being conducted in other countries, Everri et al. can provide additional information to discuss and compare with the results of this paper.

Response: Thank you very much for your kind and valuable suggestions. We have cited this article and made relevant modifications to the Introduction and Discussion sections. In the Introduction, this publication supports the argument regarding the impact of COVID-19 lockdown on caregivers’ well-being. In the Discussion, the discussion on the link between multimedia usage and parenting anxiety in this study provided strong supports for our viewpoint. Please refer to citation [16] for more details.

Question 2: Aim and research objectives should be specified. The way in which goals are presented fit a qualitative methodology, while this is a quantitative paper in which it is expected that authors have hypotheses and assumptions about the relationship among the considered variables.

Response: Thank you. Combined with other reviewers’ comments, we have revised the title using a more precise statement. Additionally, we have made modifications to ensure a more comprehensive and accurate representation of our research hypothesis and objectives, as follows: “Considering the special features of young children and lack of localized evidence on the special stage of the lockdown in Shanghai, we hypothesized that parental anxiety, pa-rental practices, and parent-child relationships among families with young children were significantly influenced by caregivers’ and children’s characteristics and factors related to the COVID-19 lockdown. Therefore, this study aimed to (1) describe the characteristics of parental anxiety, parental practices, and parent-child relationships; (2) clarify the influence of children's characteristics, family features, and experiences related to the pandemic on parental anxiety, parental behaviors, and parent-child relationships; and (3) explore the changes in parental practices among families with young children during the COVID-19 lockdown in Shanghai, China. This study was intended to provide evidence on minimizing the negative effects of pandemic prevention and control policies on families with young children from the perspective of parental characteristics.” Please refer to pp. 2-3, “1. Introduction”.

Question 3: Additionally, they should provide more details about the model that they planned to test.

Response: Combined with other reviewers’ suggestions, we have provided additional details on the specific measurement methods and variable characteristics of the dependent variables, as follows: “In terms of the dependent variable of parental anxiety, it was measured on a scale ranging from “none” to “extremely high” and determined as an ordinal variable. Considering parental practices, based on the survey results, the monthly frequency of parent-child activities was assigned a value from 0 (never) to 6 (everyday). The difference between the values pre- and post-pandemic was then calculated as the dependent variable. Parent-child relationships were measured and assigned a value based on a 6-point warmth and invasiveness scale. Considering the dependent variable of mothers’ perceptions of their warmth towards their children, it was classified into three ordinal levels (0–11, 12–19, 20–35); the dependent variable of mothers’ perceptions of their invasiveness towards their children was also divided into three ordinal levels (0–12, 13–17, 18–35). Additionally, principal component analysis was conducted using the regression scores approach to address the multicollinearity. This method allowed the composite variable to reflect the severity of parenting issues and external parenting difficulties through a numerical magnitude [30].” Please refer to pp. 4, “2.3. Statistical Analysis”.

Furthermore, we have included more comprehensive information regarding the regression model, as follows: “A univariate analysis of significant changes in the frequencies of parent-child activi-ties was conducted. Changes in the frequency of reading books or looking at picture books with children were significantly influenced by caregivers’ occupations, family residence, children’s age, whether caregivers worked from home, and whether caregivers were anx-ious about children’s physical development (Table S1). The above-mentioned variables were included in the regression analysis.” “Changes in the frequency of telling stories to one’s child were significantly influenced by caregivers’ age, caregivers’ occupations, family residence, whether caregivers worked from home, and anxiety about the child’s physical development (Table S1). The above-mentioned variables were included in the regression analysis.” Please refer to pp. 9, “3.4. Changes in parental practices and its influencing factors”.

Question 4: please reference the scales used or specify if the scales were specifically developed for the study.

Response: Apologize for the neglect of these references. We have provided proper citations for the sources of the 5-Point Likert Scale, MICS6, and MORS-SF, as indicated in citation [29], [30], and [31], respectively.

Question 5: The presentation of results would benefit from a better organization of the material. This might be due to the lack of clarity in the study hypothesis and a clearer description of the considered variables.

Response: Thank you. The study hypothesis has been added as follows: “Considering the special features of young children and lack of localized evidence on the special stage of the lockdown in Shanghai, we hypothesized that parental anxiety, pa-rental practices, and parent-child relationships among families with young children were significantly influenced by caregivers’ and children’s characteristics and factors related to the COVID-19 lockdown.” Additionally, the corresponding research objectives were revised as follows: “Therefore, this study aimed to (1) describe the characteristics of parental anxiety, parental practices, and parent-child relationships; (2) clarify the influence of children's characteristics, family features, and experiences related to the pandemic on parental anxiety, parental behaviors, and parent-child relationships; and (3) explore the changes in parental practices among families with young children during the COVID-19 lockdown in Shanghai, China. This study was intended to provide evidence on minimizing the negative effects of pandemic prevention and control policies on families with young children from the perspective of parental characteristics.” Please refer to pp. 2-3, “Introduction”.

Further, we correspondingly adjusted the subheadings in the Results section to better present the results, such as “3.4. Changes in parental practices and its influencing factors”.

Question 6: There is reference to Table S1 that should be inserted.

Response: Due to the length of Table S1 and considering its impact on reader experience and publishing convenience, we have complied with the submission requirements and placed it in the supplementary materials. Please refer to Table S1 in the supplementary files for further details.

Question 7: I suggest that authors replace ‘basic characteristics’ with ‘demographic characteristics’.

Response: Thank you. Considering that Table 1 included not only demographic characteristics but also parenting and lockdown-related features, we revised the title as "Demographic, Parental, and Lockdown-Related Characteristics" for more accurate statement. Please refer to pp. 5, “3.1. Demographic, parental, and lockdown-related characteristics of caregivers, families, and children”.

Question 8: Overall, the paper requires major revisions. Perhaps authors should consider reducing the number of variables considered and develop stronger hypothesis. The paper lacks in consistency and this reduce the potential strengths of the results.

Response: Thank you for your valuable comments. Based on the revised hypothesis and research objectives, we made comprehensive revisions on the title, Introduction, Methods, and Results to improve the overall coherence. Further, combined with other reviewers’ suggestions, we made more revisions to stimulate the potential strengths of this study. Please refer to the whole revised manuscript.

Reviewer 3 Report

This capable study examines a Chinese family factors related to COVID. It could be improved with attention to the following. 

1. Was parental religiosity examined? See Bartkowski’s research on religion and child development as well as religion and COVID. If not measured, then this could be a direction for future research.

2. An additional conceptual framework would be welcome. What about a perspective that conceptualizes the family as situated within a broader ecological system? Ecological systems theory would be an excellent addition.

3. I’d suggest a more explicit and detailed recommendation for follow-up qualitative research through in-depth interviews and ethnographic research. 

Well done!

Author Response

Question 1: Was parental religiosity examined? See Bartkowski’s research on religion and child development as well as religion and COVID. If not measured, then this could be a direction for future research.

Response: Thank you for your kind suggestions. We do agree the importance of religion. However, most people in China, especially in Shanghai, do not have explicit religious beliefs, nor do they engage in corresponding religious practices or activities. As a result, we had not considered religious beliefs in this study. However, we agree that it should be seriously considered in the future study, which has been stated as a limitation: “Fourth, some family factors, such as religion, were not measured. Face-to-face in-depth interviews should be conducted to yield better results.” Please refer to pp. 12, “4. Discussion”.

Question 2: An additional conceptual framework would be welcome. What about a perspective that conceptualizes the family as situated within a broader ecological system? Ecological systems theory would be an excellent addition.

Response: Thank you. The Ecosystem Theory do help improve our study design and background. We have added this theory to better explain the importance and conceptual framework of this study, as follows: “Based on the Nurturing Care Framework [7], responsive caregiving theory [8], and ecosystem theory [9], interactions between caregivers and children are closely related with children’s developmental health, especially in the early stage of life; such interactions are mainly influenced by caregivers’ parental characteristics in the home environment [10]. Specifically, ecosystem theory posits that caregivers’ behaviors and parent-child relationships constitute the most crucial connections within the entire system [11], and the physical and mental health of caregivers constitutes the secondary links [12]. Therefore, care-givers’ parental characteristics are typically reflected as caregivers’ mental health, behaviors, and relationships with their children [13].” Please refer to pp. 2, “1. Introduction”.

Question 3: I’d suggest a more explicit and detailed recommendation for follow-up qualitative research through in-depth interviews and ethnographic research.

Response: Due to the restrictions imposed by the pandemic lockdown measures, we were unable to conduct in-person, face-to-face in-depth interviews with the participants. However, we agree that face-to-face in-depth interviews are valuable in enhancing data accuracy and reducing potential information bias, which should be conducted in future studies. It has been added as a limitation, please refer to pp. 12, “4. Discussion”.

Round 2

Reviewer 3 Report

I commend the authors on an impressive revision. As I see it, after some additional copyediting, the paper is ready for publication. 

Some additional proofreading is recommended.